# Investigation of a Cabin Suspended and Articulated Rescue Vessel in Terms of Motion Reduction

**Jiong Li, Xuesong Bai, Yang Li, Hongwang Du** **, Fangtao Fan, Shuaixian Li, Zhi Li and Wei Xiong ***

School of Marine Engineering, Dalian Maritime University, Dalian 116026, China
* Correspondence: xiongwei@dlmu.edu.cn

**Abstract:** To solve the problem that small rescue vessels are seriously affected by waves when working at sea, which causes damage to the crew and equipment, a catamaran with a suspended and articulated cabin is designed. The multi-body dynamic model of the whole vessel based on SimMechanics is built, and the prototype is tested on the water. The correctness of the model is verified based on the three aspects of the roll motion, pitch motion, and heave motion of the rescue platform. The motion of the rescue platform is simulated under the conditions of bow waves and large heave waves, and the RMS value of the vertical acceleration of the rescue platform is taken as the response target to analyze the influence of the suspension system parameters of the whole vessel. The suspension system has good buffering and motion reduction effects under various working conditions, and it has an obvious reduction effect on the vertical acceleration at the rescue platform. The larger the amplitude of the wave impact, the more obvious the effect. The articulated system reduces the roll angle and the heave amplitude under the conditions of bow waves.

**Keywords:** rescue vessels; suspension system; articulation system; parameter optimization; dynamics simulation; experimental verification

## 1. Introduction

Small emergency rescue vessels play a huge role in offshore life rescues [1,2]. However, the existing small emergency rescue vessels generally have the problem of poor seakeeping, so it is urgent to develop a type of life rescue equipment that can operate in rough sea conditions. A suspended and articulated cabin catamaran rescue vessel with an added suspension system between the main cabin and the two monohulls to buffer and absorb the partial motion impact of the two monohulls is proposed to improve the seakeeping ability of the main hull and enhance the existing emergency life rescue ability at sea. In this research, SimMechanics is used to establish the dynamics model of a catamaran rescue vessel with a suspended and articulated cabin, and the water test is used to verify the correctness of the dynamics model. Through the iterative simulation of the model, the selection method of the key parameters of the suspension system is discussed. Finally, the effect of the suspended system and the articulated system on the motion reduction effect under different sea conditions is discussed by comparing the standard criterion of seakeeping.

In the research field of small-sized emergency rescue boats, Europe and the USA began early and have invented many types of developed rescue ships. For example, the 47-foot Motor Lifeboat, a lifeboat manufactured by the TEXTRON Company and recognized as the standard lifeboat of the United States Coast Guard, has been widely equipped across the world [3]. At present, China has introduced the well-known foreign rescue ships 20 ARUN and TYNE-class fast rescue boats called the "Huaying" series to the Royal National Lifeboat Institution of Britain [4]. However, China has put more effort into the optimization of hull structures and research on professional rescue equipment. The Shenzhen HiSiBi boat technology development Co., Ltd., has been working together with the Dalian Maritime

University, Harbin Institute of Technology, and other academic institutions in the field of design and construction technology for self-righting boats since 2003. After going through the stages of design research, boat construction testing, and performance evaluation, the project was finally recognized by expert technical appraisal in October 2007 [5]. However, the common problem of the poor seakeeping capability of small rescue boats has never been solved.

In the field of ship and ocean engineering, the following three methods are generally adopted to improve the seakeeping performance of marine equipment: installing bionic fins, creating moving mass, or building multi-body structures. (1) A bionic fin is installed to increase the oscillation damping of marine equipment to improve the anti-rolling moment. Bilge keels, fin stabilizers, and rudder stabilizers are the widely accepted devices in this category. Bilge keels are simple [6]. Fin stabilizers and rudder stabilizers are both categorized as active damping devices, and fin stabilizers are the best active stabilizers at present. They can keep the rolling amplitude within 3° in any sea condition. The world's first fin stabilizer was designed in 1923 by Dr. Motora from Mitsubishi Heavy Industries. Due to their complex structure and relatively high costs, fin stabilizers are generally applied to large ships and warcraft [7]. Since a yaw moment and a roll moment are generated when the rudder is operating, it is certainly a logical idea to exploit this effect for roll reduction, that is to say, controlling the course and reducing the roll at the same time. Rudder stabilizers were first proposed in 1972 and were successfully used on a merchant ship [8]. Because rudder stabilizers require large amounts of power and large rudder speeds, the steering gears of civil ships must be modified before the installation of these gears. In addition, the rudder stabilizer controller is highly sensitive to ship parameters because slight changes in the ship structure, ship loading, ship speed, and steering gear parameters will reduce the anti-rolling effect and even make the anti-rolling control fail [9]. (2) The second method involves shifting a certain quantity of mass to generate an anti-rolling moment. Tank stabilizers, weight stabilizers, and gyro-stabilizers all adopt this principle. A tank stabilizer is one kind of special water tank installed in a hull. When a ship is rolling, the water in the tank can flow from one side to the other side, thus creating a stable moment to resist rolling. A U-type water tank and a flume tank are among the most commonly used water tanks, and they are categorized as passive anti-rolling water tanks. It is necessary to adjust the natural frequency of oscillation in the designed water tank to be equal to the natural frequency of ship roll to achieve the so-called "double resonance anti-rolling principle". In the case of resonance, the direction of the stabilizing torque caused by the weight of water in the water tank is exactly opposite to the direction of the wave rolling torque to reduce the rolling in the resonance region [10]. The active tank stabilizers are equipped with a throttle valve in the water tank channel, and by adopting different control theories, people can adjust the valve's opening and closing degree so that the tank stabilizers can work effectively in a wider frequency range than passive water tanks can [11]. Changing the mass of the liquid into that of a solid is the principle of the weight stabilizer, and the mass block can also move up and down with a response speed that is much faster than that of the liquid [12]. A gyro-stabilizer utilizes the rotation force (rotation moment) of the gyro in the opposite direction to the rolling of the ship, thus inhibiting the rolling [13]. In recent years, gyro-stabilizers have had relatively mature development, and their anti-rolling effect is generally 33–47%. Because gyro-stabilizers are convenient to install and have low noise and no outboard parts, they are widely applied in small yachts [14]. Globally, the major manufacturers of gyro-stabilizers are Seakeeper from the United States, Halcyon and SEA GYRO from Australia, Mitsubishi Heavy Industries from Japan, and Shanghai YuYi Marine Equipment Co. Ltd. and Shanghai JiWu Technology Co. Ltd. from China are the main manufacturers. (3) The third method involves utilizing a multi-body structure to isolate wave disturbances. This method has been widely studied in recent years. As a result, many theories and innovative designs have been put forward. The general structure of traditional ships is mostly a single rigid body or demihulls fixed together, but a ship adopting this design method is composed of multiple parts, which provide multiple degrees of freedom

for the structure [15]. This type of ship is usually composed of the main cabin, the connecting mechanism, and the lower demihulls. The connecting mechanism has the flexibility to stretch, expand, and rotate so that it can isolate the movement between the main cabin and the lower demihulls. The connecting mechanism is generally composed of spring and damper suspension components, a hydraulic cylinder, or other actuators. In summary, each of these three methods for improving the seakeeping performance of ships has its own advantages and disadvantages. The first two methods have been relatively developed, and now the main focus is on device optimization and advanced control algorithms. The third method is still under study, and experts are mainly focusing on the prototype experiment and model validation.

In 1987, Japan's Mitsubishi Heavy Industries built a catamaran, the HSCC (Hi-Stable Cabin Craft), with automatic stabilizing hydraulic actuators. The computer on board collected the cabin motion signals through the sensors and controlled the hydraulic cylinder promptly to keep the cabin stable. The sea test results showed that the cabin motion amplitude was only one-third of that of the lower Demihulls [16]. In 1990, a full-size catamaran, namely the HSCC Voyager, with a total length of 26.5 m and a capacity of 200 passengers, was built and tested at sea. The relative surge, sway, heave, and yaw motions between the cabin and the demihulls were restrained by a gimbal device. The pitch and roll motions of the cabin were counteracted by several hydraulic cylinders driven by a control system. About 75% of the roll and pitch of the cabin was reduced compared with that of the demihulls [17]. In both the HSCC and the HSCC Voyager, Mitsubishi Heavy Industries' design idea was limited by the addition of a large and complete mechanical suspension system into a limited hull space. Its safety, stability, and economy limit further development of this solution. In 2007, Proteus, the first product of Marine Advanced Research in the United States, took its maiden voyage in San Francisco Bay. Two flexible pontoons were connected by a large front arch, and the special wave adaptive module effectively reduced the ship's oscillation caused by waves, similar to a giant spider walking on water [18]. Later, the institute launched 12-foot and 33-foot verification ships as technology demonstrators and cooperated with Virginia Tech and The University of Iowa to conduct in-depth research on a multi-body dynamics model and a hydrodynamic model [19]. Instead of a complex mechanical system to isolate the movement between the main hull and the demihulls, Proteus, and subsequent generations used four passive suspensions to achieve this goal. Although this design was simple and effective, it had resonance phenomena under certain sea states, which need further improvement by the designer. Virginia Tech's multi-body dynamics study of this design adopted more automotive field methods [20]. Based on their works, this study proposes a four-degree-of-freedom multi-body dynamics model and discusses it based on the analysis methods in the field of marine engineering. David Hall, an electrical engineer, tested a multihull catamaran in 2012. The ship was equipped with real-time electronic pneumatics, which consisted of state-of-art actuators and air suspensions. The low-level oscillation of the cabin at a high forward speed was observed in a sea trial [21]. David Hall's design idea was based on very precise pneumatic components and a reliable real-time control system, furthermore, the load capacity of the design was not clearly stated. Considering the stability of pneumatic components under the condition of large load impact, the design needs to be further studied on the basic components. In 2014, Nauti-Craft Pty Ltd. in Australia developed a passive reactive interlinked hydraulic system that was implemented on a cabin-suspended catamaran. The sea trial of the full-scale ship demonstrated a high level of ride comfort of the suspended cabin [22]. The design of Nauti-Craft Pty Ltd. had been validated by numerous experiments, and the entire hydraulic system had both active and passive suspension solutions. However, the system still does not solve the problem of the entire suspension taking up too much space. Although these unique design schemes have improved the seakeeping capacity of small ships, they have also added active actuating components or passive damping structures to the connection mechanism, causing the overall mechanism to be relatively complex and to have difficulty fully utilizing its

advantages in rough sea conditions. The catamaran with suspended and articulated cabin developed by Prof. Xiong Wei's team at Dalian Maritime University has been proven to be able to buffer the load caused by wave impact and ensure the smoothness of the rescue platform through prototype test in a wave pool [23]. Additionally, the rescue vessel did not use complex pneumatic or hydraulic components, the whole suspension system only took up a small part of space.

In view of utilizing a multi-body structure to isolate wave disturbances in marine engineering, most of the research focused on the development of the prototype, however, the investigation of the model and the optimization of key parameters were not clear enough. In this research, a multihull dynamics model is established for a rescue vessel with a suspended and articulated cabin, and water experiments are conducted in a wave tank to verify the rationality and accuracy of the modeling.

## 2. Structure of the Rescue Vessel

### 2.1. Overall Structure of Rescue Vessel

The rescue vessel consists of a rescue platform, a front arch, a suspension system, two pontoons, a rear arch, and an engine pod, as shown in Figure 1. Between the front arch and the pontoon, a suspension system is added. The front arch is connected to the rescue platform and the suspension system by spherical joints, and the rear arch is connected to the left and right pontoons by articulation joints with rotational degrees of freedom in the horizontal and vertical planes, which constitute the articulated system of the vessel.

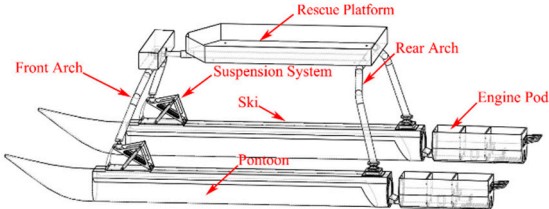

**Figure 1.** Structure of the rescue vessel.

According to the application and rescue environment of the rescue vessel [24], the specific dimensions of the rescue vessel designed in this research are shown in Table 1.

**Table 1.** Structure size of the designed rescue vessel.

| Symbol | Parameter | Dimensions |
|--------|-----------|------------|
| $L$ | Overall length | 3.8 m |
| $L_w$ | Length of water plane | 3.5 m |
| $B$ | Beam | 1.88 m |
| $B_h$ | Hull diameter | 0.25 m |
| $W$ | Hull space | 1.63 m |
| $\nabla$ | Displacement | 150–220 kg |
| $D$ | Average draft | 0.11–0.16 m |
| $m_1$ | Mass of the floating body | 53.5 kg |
| $m_2$ | Mass of the cabin | 43 kg |
| $I$ | Moment of inertia of the cabin | 22.25 kg·m$^2$ |
| $L$ | Overall length | 3.8 m |

### 2.2. Design Characteristics of the Rescue Vessel

2.2.1. Suspension System

The novel idea of the suspension system mainly comes from the automobile industry. The suspension system of an automobile can effectively attenuate vibration by reducing the amplitude, thus effectively improving the comfort and handling stability of the automobile while driving [25].

The suspension system of the rescue vessel designed in this research is mounted on the left and right demihulls, and this system is mainly used to connect the front arch and the rigid ski and transfer the impact load of the wave acting on the pontoon, the propulsion force of the propulsion system, and the steering torque to the front arch to ensure the normal navigation of the vessel. The suspension system, as shown in Figure 2, is mainly composed of a damper on the left side, a leaf spring at the end, and a rocker arm connecting the two. The damping coefficient of the damper and the stiffness coefficient of the leaf spring in the suspension system are the most important factors [26]. When the rescue vessel travels in rough sea conditions, the vertical motion of the pontoon caused by the sea waves is attenuated and buffered by the damper and the leaf spring of the suspension system, so the amplitude of the impact is rapidly reduced and the frequency is mitigated, thus improving the stability of the rescue platform and reducing the seasickness of the driver and the rescued personnel.

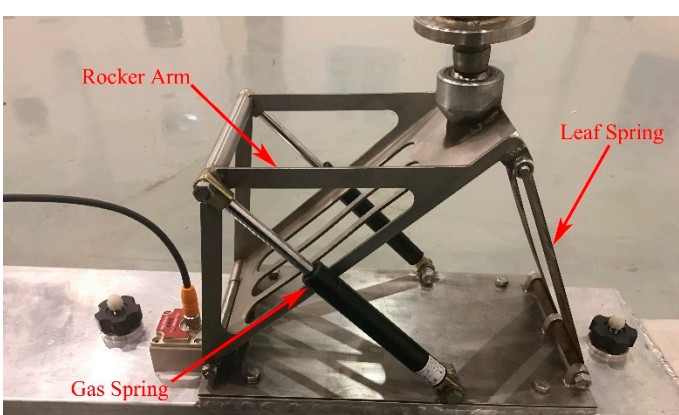

**Figure 2.** Structure of the suspension system.

### 2.2.2. Multi-Degree-of-Freedom Articulated System

In contrast to traditional vessels, although the vessel has two demihulls contacting with the water surface, the strong frame connecting the two demihulls is not a whole rigid body structure, but rather is articulated by multiple rigid body components, so the seakeeping performance of the vessel is affected not only by the key parameters of demihulls but also by the multi-body dynamic of the articulated structure shown in Figure 3. The whole mechanism is connected by three spherical joints, which are located between the front arch and the main hull, and between the front arch and the left and right suspension mechanisms. The spherical joint connections only limit the displacement motion in three directions and do not limit the rotation motion. When one side of the demihull encounters the sea wave, the other demihull can adapt to the undulating motion of the sea wave instead of the intense collision of the traditional vessel. The rear arch sections are rigidly attached to the rescue platform but are allowed to rotate relative to each pontoon in pitch [27].

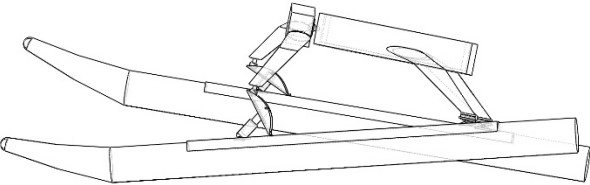

**Figure 3.** Articulated structure of multi-degree of freedom.

### 2.3. Kinematic Description of the System

In the actual use of rescue vessels, heave motion and roll motion have the greatest influence on the seakeeping performance, so the motions of the pontoons and roll motion are added to form the four-DOF model shown in Figure 4, based on the single-DOF model

proposed by Manhar R. Dhanak [28]. $k_1$ and $k_2$ are the equivalent stiffness coefficients of the flexible pontoon and suspension system. $\mu$ is the equivalent damping coefficient of the suspension system. $w_l$ and $w_r$ are the displacement excitations of the waves for the left and right pontoons in the vertical direction, respectively, $z_{1l}$ and $z_{1r}$ are the displacements of the left and right pontoons in the vertical direction, respectively, $z_2$ is the displacement in the vertical direction of the center of mass of the rescue platform, and $\theta$ is the roll angle of the rescue platform.

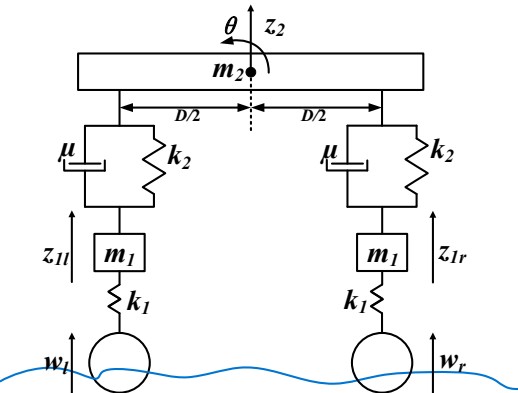

**Figure 4.** Four degrees of freedom model diagram.

The multi-body dynamic model of the rescue vessel is depicted in Figure 4, and the coordinate system follows the right-hand rule. The Lagrange method is used to establish the following differential equations of motion:

$$\begin{cases} m_1 z''_{1f} = -k_{s1}(z_{1f} - w_{1f}) + k_{s2}(z_0 - \frac{D}{2}\theta - z_{1f}) + \mu_d(z'_0 - \frac{D}{2}\theta' - z'_{1f}) \\ m_1 z''_{1r} = -k_{s1}(z_{1r} - w_{1r}) + k_{s2}(z_0 + \frac{D}{2}\theta - z_{1r}) + \mu_d(z'_0 + \frac{D}{2}\theta' - z'_{1r}) \\ I\theta'' = \frac{D}{2}k_{s2}(z_{1r} - z_{1f} - D\theta) - \frac{D}{2}\mu_d(2z'_0 - z'_{1f} - z'_{1r}) \\ m_2 z''_0 = -k_{s2}(2z_0 - z_{1f} - z_{1r}) - \mu_d(2z'_0 - z'_{1f} - z'_{1r}) \end{cases}, \quad (1)$$

the heave motion and roll motion of the rescue platform can be solved by Equation (1).

The displacement and the hydrodynamic coefficient of the pontoon in waves can be theoretically calculated with the new strip method or experimentally estimated by conducting a slamming test [29]. However, these two measurements have some properties that are not easily defined or solved mathematically, such as the damping coefficient and the added mass of the pontoons in the water, which change as a function of the frequency.

## 3. Modeling and Experimental Verification of Rescue Vessel

### 3.1. Whole Vessel Dynamics Model Based on SimMechanics

SimMechanics in a Matlab environment is used for the multi-body dynamic modeling, and the three-dimensional model is imported into SimMechanics from Solidworks [30]. All of the components are regarded as rigid bodies without deformation, and the friction between spherical joints and revolute pairs is not considered [31]. Figure 5 shows the degrees of freedom and constraints of the whole vessel model. The kinematic joints between parts are mainly divided into three types, namely spherical joints, prismatic joints, and revolute joints. The rescue platform is connected to the front arch by a spherical joint, and the front arch and the upper end of the rocker's arm are similarly connected by spherical joints. The spherical joint constrains the translation along the X, Y, and Z axes and retains the rotation around the X, Y, and Z axes. The lower end of the rocker's arm is connected to the pontoon by a pin shaft, and a revolute joint is arranged in the model, which restrains the translation along the X, Y, and Z axes as well as the rotation around the X and Z axes and retains the rotation around the Y axis. The pontoon and the rear arch are connected by a hinge, which is also set as a revolute joint in the model, which constrains

the translation along the X, Y, and Z axes and the rotation around the X axis and retains the rotation around the Y and Z axes. The upper end and the lower end of the shock absorber are considered to be prismatic joints, which constrain the translation along the X and Y axes as well as the rotation around the X, Y, and Z axes and retain the translation of the Z axis.

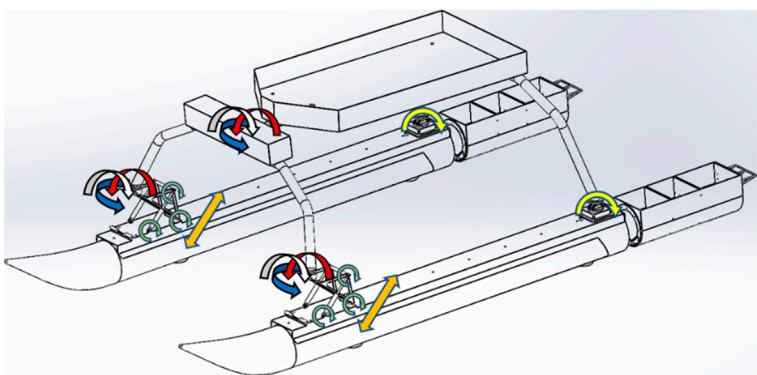

**Figure 5.** Degrees of freedom and constraints of the rescue vessel.

In a time domain simulation, the states of each rigid body can be classified by their respective positions and velocities [32]. The determined displacement or velocity is input from the signal input module so that the pontoon has a certain motion. Then the suspension system is squeezed or stretched, and the impact is finally transmitted to the rescue platform through the buffering and absorption of the suspension system. The whole vessel model is shown in Figure 6, in which parameters such as the mass and moment of inertia of the parts are calculated according to the model. The motion responses of the rescue platform are detected by the Transform Sensor, and these responses are the heave motion, the pitch motion, and the roll.

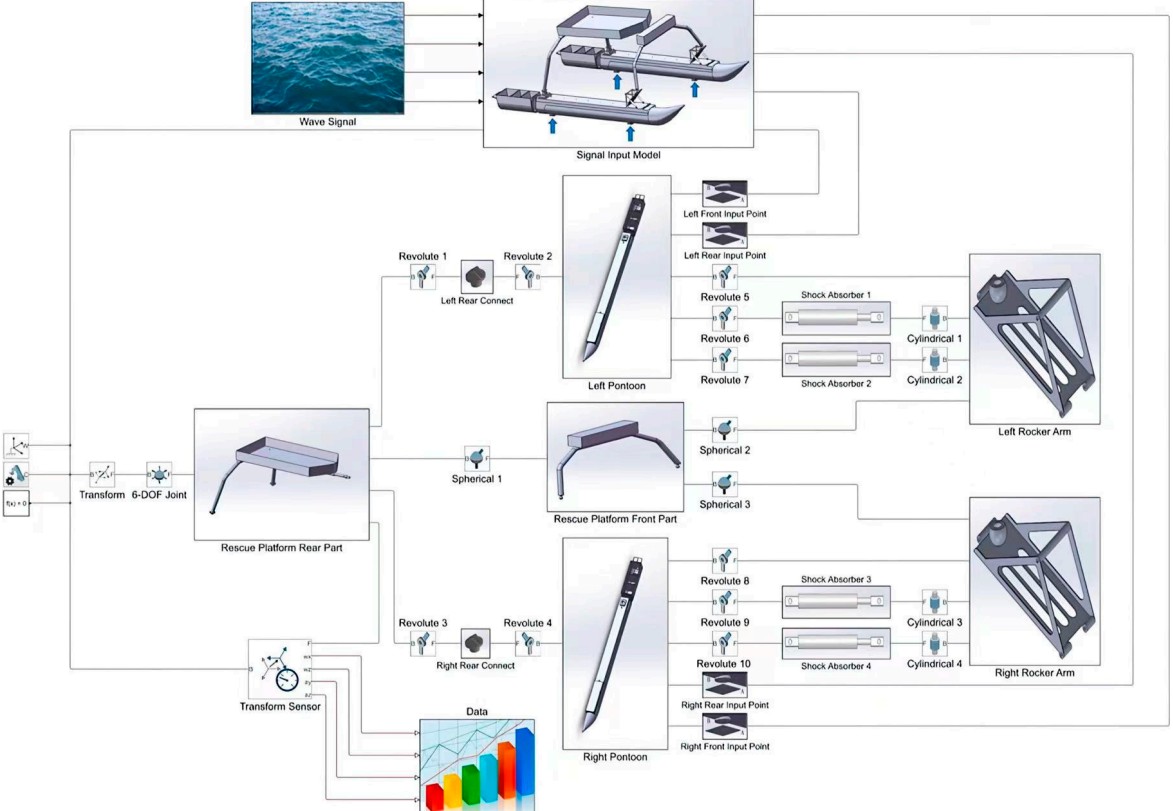

**Figure 6.** Block diagram model of the rescue vessel.

### 3.2. Experimental Verification

3.2.1. Experiment on Water

The verification experiment of the rescue vessel is carried out in the Rescue and Salvage Engineering Laboratory of the Dalian Maritime University. The wave tank is 50 m long, 30 m wide, and 6 m deep. The wave tank can simulate moderate waves, medium waves, and hard waves with average wave heights of 0.42 m, 1.02 m, and 1.60 m, respectively. The experiments are conducted with several incident wave directions. The movement data for the pontoon and the rescue platform are collected for different kinds of waves, as shown in Figure 7.

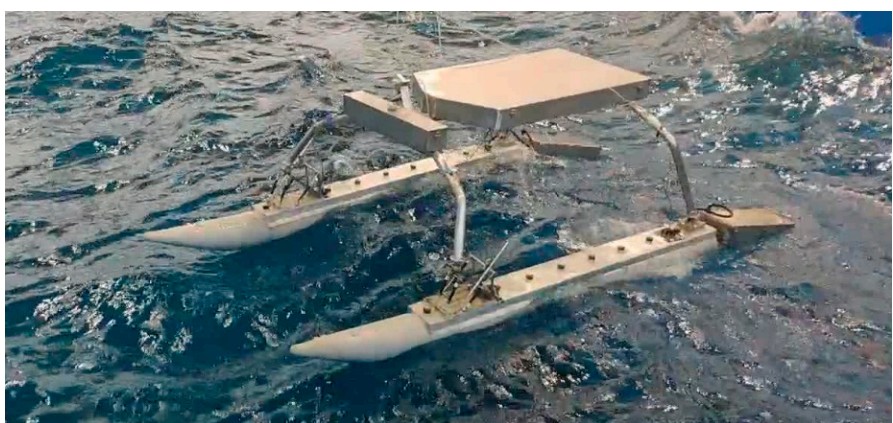

**Figure 7.** Photo of the rescue vessel during the experiment.

The installation position of the sensor in the experiment is shown in Figure 8. The 16 g accelerometers are oriented vertically and mounted on top of the pontoon skis to measure the vertical acceleration data. This is consistent with the position of the excitation point in the simulation model. A single tri-axial 16 g gyroscope is mounted on the sprung mass close to the center of mass in the rescue platform to measure the vertical acceleration, roll, and pitch angles with the position of the motion measurement point in the simulation model. An Advantech USB-4711a acquisition card is selected as the data acquisition card.

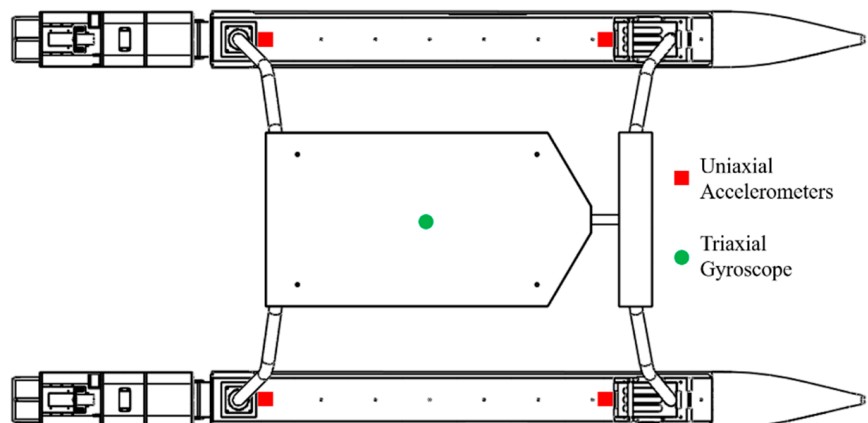

**Figure 8.** Diagram of the sensor arrangement.

3.2.2. Data Pre-Processing

Before the measured accelerometer data can be used as inputs of the simulation model, first the data have to be transformed into global vertical displacements. Transforming the data requires two separate integration procedures. Due to the limitations of the frequency domain integration, integrations are performed in the time domain. Integration in the time domain is more accurate at lower frequencies to analyze the dynamics of the rescue

vessel [33]. To obtain usable outputs from the data, the data need to be preprocessed before integration to minimize potential errors. To remove the offset due to gravity from the raw data, the average acceleration of the data set is calculated and subtracted from each data value to detrend the data. The high-frequency engine noise is removed from the data through the use of a low-pass filter, set at 30 Hz to avoid interference occurring at lower frequencies of the ocean inputs. Finally, a high-pass filter with a low cutoff frequency of 0.35 Hz is used to eliminate small artifacts in the data that can lead to moving means once integrated [34]. A zero-phase filtering approach is implemented to eliminate any temporal shifts due to the filtering scheme [35]. The filtered accelerometer data are available to perform integration operations. A cumulative trapezoidal integration scheme is used to generate the velocity data. The velocity data are processed again using the same zero-phase filter and trapezoidal integration scheme to obtain the global displacement data as a function of time. The entire process of converting the raw acceleration data into usable displacements is shown in Figure 9.

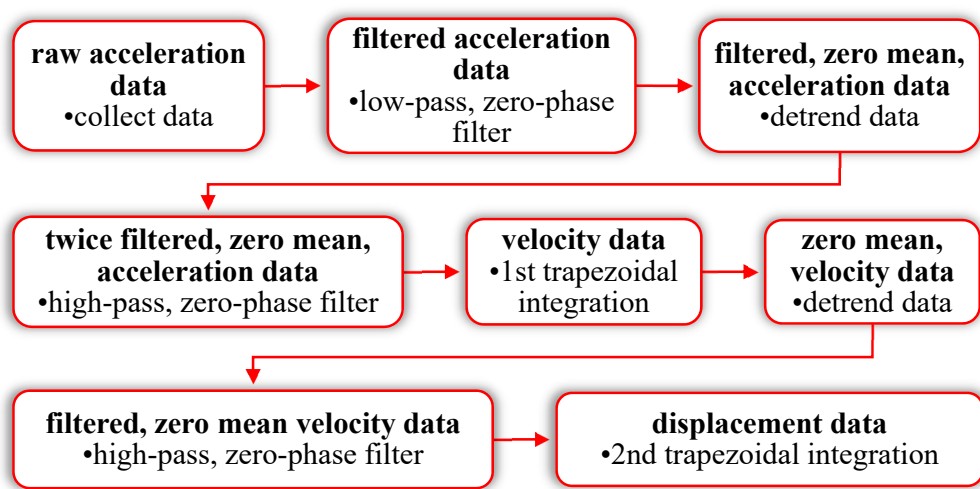

**Figure 9.** Data integration procedure.

### 3.2.3. Model Validation

The validation of the multi-body dynamic model is a crucial step in its application to dynamic prediction algorithms. A direct comparison is made between the simulated rescue vessel movement outputs and the experimentally recorded movement data from testing. To be considered suitable, the multi-body dynamic model must be able to predict vessel motions above the pontoons with reasonable accuracy. The heave displacement, roll angle, and pitch angle of the rescue platform in the model are calculated and compared with the data measured in the experiment. The comparison of the simulation data and the experimental data is shown in Figure 10. It can be seen from the figure that the amplitude and phase are in good agreement. Specifically, the mean absolute error of the heave displacement in simulation data and the measured data is 0.01 m. In addition, the mean absolute errors of the roll angle and the pitch angle in the simulation data and the measured data are 1.12° and 1.82°, respectively. The simulation data are very close to the experimental data, which shows that the model is correct and reliable, and the dynamic model of the rescue vessel can be further studied.

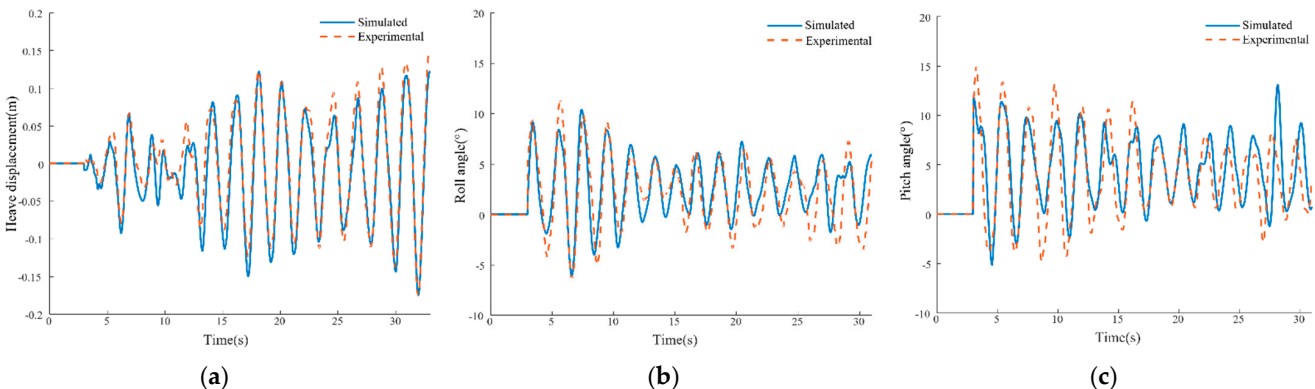

**Figure 10.** Comparison of simulated and experimental data. (**a**) Heave displacement; (**b**) Roll angle; (**c**) Pitch angle.

## 4. Study of Key Parameters

Referring to the idea of a vehicle vibration test bench, the motion of the rescue vessel is simulated with the impact of waves using the SimMechanics model, and then the influences of the stiffness coefficient and the damping coefficient of the suspension system in different working conditions are analyzed [36]. The initial parameters are as follows. The stiffness of the suspension system is 4500 N·m$^{-1}$ and the damping ratio is 0.6; that is, the damping coefficient of a single suspension system is 294 Ns·m$^{-1}$. During voyages and operations at sea, vessels usually encounter beam waves, head waves, bow waves, and large heave motion [37], among which bow waves can be regarded as the combination of head waves and beam waves. Hence, the SimMechanics model is used to simulate the rescue vessel during bow waves and large heave motion, and the dynamic characteristics of the suspension system are analyzed. The input frequency of wave impact is 0.8 Hz, and the amplitude refers to the displacements of the experimental data at the pontoon, which are 0.1 m, 0.25 m, and 0.4 m. The input method and the animation are shown in Figure 11.

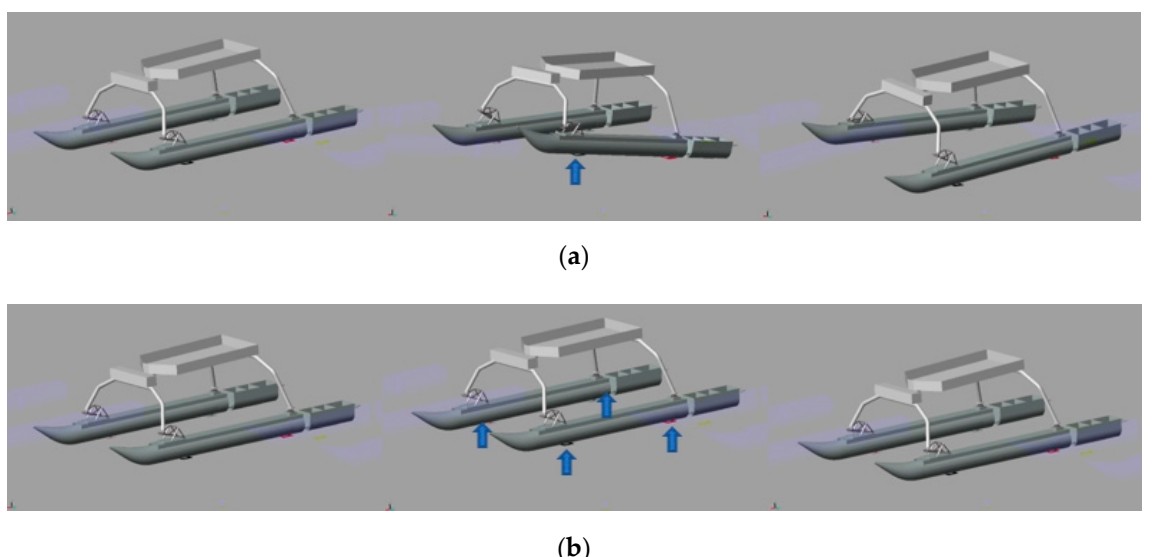

**Figure 11.** Simulation animation screenshots with different wave impacts. (**a**) Simulation with bow waves; (**b**) Simulation with large heave motion.

The root mean squared value of the vertical acceleration at the mass center of the rescue platform is selected as the evaluation index. For the cases of wave impact in different wave directions, the original stiffness coefficient and the damping coefficient are increased and decreased by 25% and 50% for the system, respectively. By changing the damping

coefficient of the suspension system and setting the damping coefficient with the minimum root mean square of the vertical acceleration of the rescue platform as the new damping coefficient, the influence of the stiffness coefficient on the motion of the rescue platform is studied by changing the value of the stiffness coefficient under the condition of ensuring the constant damping ratio.

### 4.1. Simulation with Large Heave Motion

The corresponding stiffness value and the damping value for the simulations are listed in Table 2, and the acceleration root mean squared value at the mass center of the rescue platform is shown in Figure 12. From the simulation results, with the different amplitudes of the wave impact, the influence of the damping change on the response of the rescue platform is consistent, and damping coefficients that are too small and too large are unfavorable to motion mitigation. When the damping value is 221 Ns·m$^{-1}$, the root mean square value of the acceleration at the mass center of the rescue platform is the minimum, and the dynamic stroke of the suspension system does not exceed the maximum stroke, so the damping value of 221 Ns·m$^{-1}$ is used as the reference value for the following stiffness coefficient study.

**Table 2.** Simulation parameters of damping value with large heave motion.

| Stiffness Coefficient (N·m$^{-1}$) | Damping Coefficient (Ns·m$^{-1}$) | Gradient of Coefficient Change |
|---|---|---|
| 4500 | 147 | −50% |
| 4500 | 221 | −25% |
| 4500 | 294 | reference value |
| 4500 | 368 | +25% |
| 4500 | 441 | +50% |

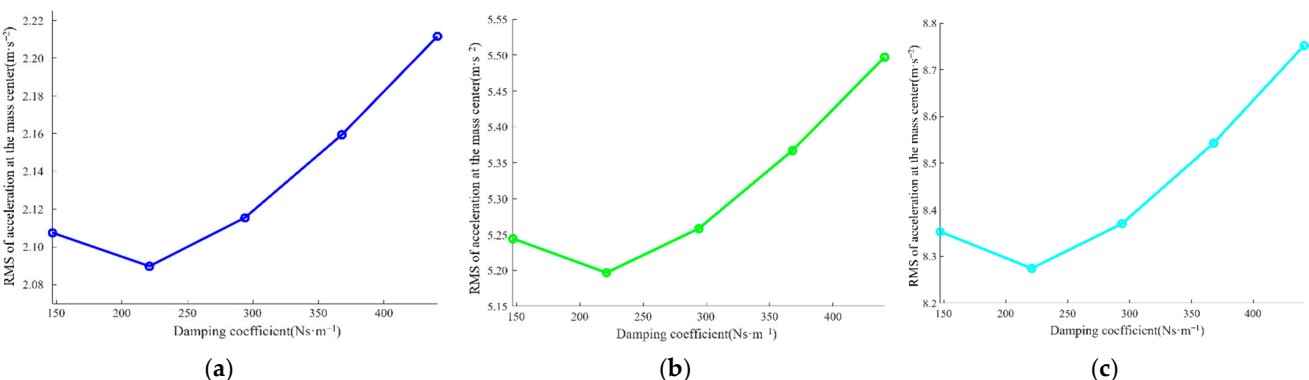

**Figure 12.** Damping change simulation comparison with large heave motion. (**a**) The amplitude of 0.1 m; (**b**) Amplitude of 0.25 m; (**c**) Amplitude of 0.4 m.

When the stiffness coefficient is 4500 N·m$^{-1}$ and the damping coefficient is 221 Ns·m$^{-1}$, the damping ratio is 0.45. For this study, the damping values are adjusted for each simulation to keep the damping ratios constant using Equation (2) for the damping ratio [38].

$$\zeta = \frac{c}{2\sqrt{km}}. \tag{2}$$

The new stiffness coefficient and damping coefficient are shown in Table 3, and the simulation results are shown in Figure 13. When the input wave amplitudes are 0.25 m and 0.4 m, a severe collision with the limit block occurs in the suspension system when the stiffness coefficient is reduced by 50%. It is noteworthy that although reductions in the suspension stiffness show a correlation with the reduced vertical accelerations, they also tend to produce larger displacements at the center of mass.

**Table 3.** Simulation parameters of stiffness value for large heave motion.

| Stiffness Coefficient (N·m$^{-1}$) | Damping Coefficient (Ns·m$^{-1}$) | Gradient of Coefficient Change |
|---|---|---|
| 2250 | 155 | −50% |
| 3375 | 190 | −25% |
| 4500 | 221 | reference value |
| 5625 | 245 | +25% |
| 6750 | 270 | +50% |

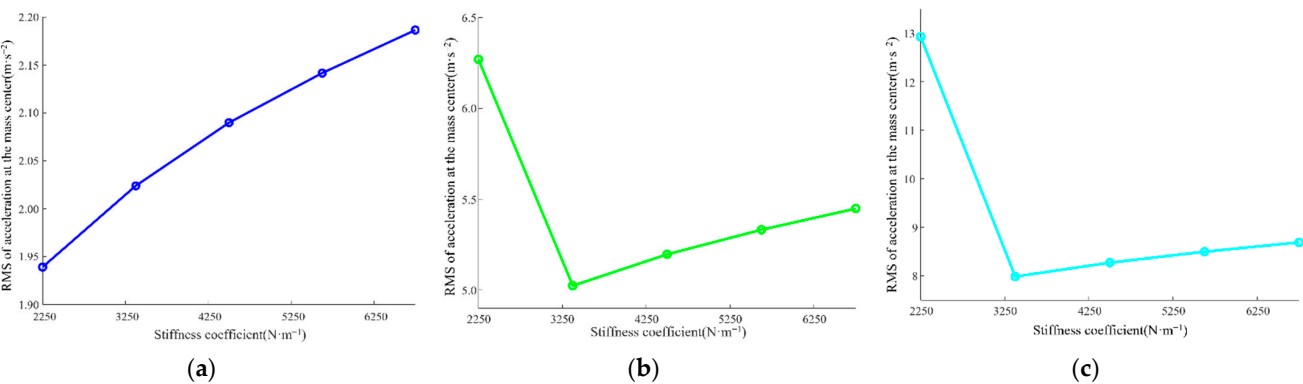

**Figure 13.** Stiffness change simulation comparison for large heave motion. (**a**) The amplitude of 0.1 m; (**b**) the amplitude of 0.25 m; (**c**) the amplitude of 0.4 m.

### 4.2. Simulation with Bow Waves

According to the simulation method of large heave motion, three new simulations are conducted and plotted against the baseline configuration described in the previous section. The simulation results are shown in Figure 14. A larger damping coefficient is more helpful to buffer and absorb vertical movement under bow wave conditions, but with the wave amplitude of 0.4 m, the dynamic stroke of the suspension system exceeds the limit position. This indicates that the reference stiffness coefficient is not enough with the input of the large amplitude value, and the ideal working effect cannot be achieved by changing only the damping coefficient. Therefore, the new reference damping coefficient is selected by increasing the damping by 50%, and only the simulation of increasing the reference stiffness is carried out. The increase gradient of the stiffness is still five gradients, with each gradient increasing by 25%.

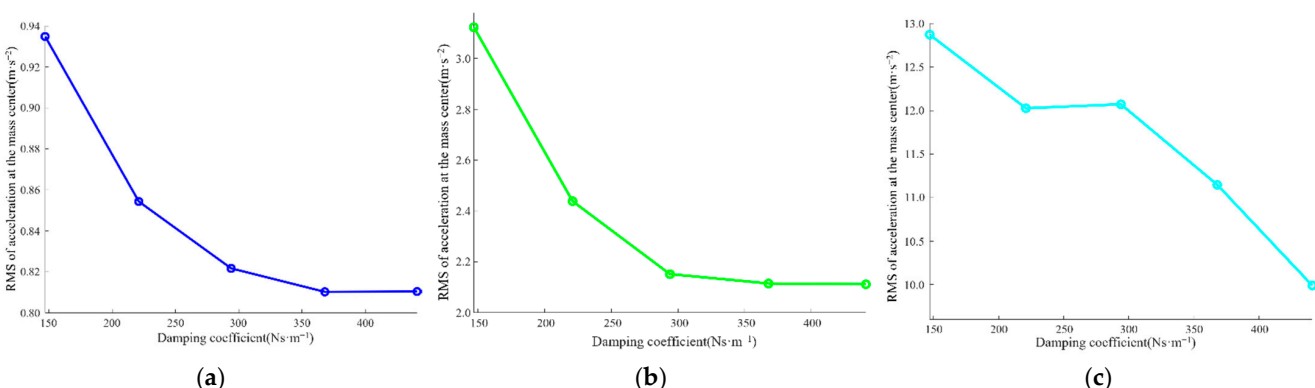

**Figure 14.** Damping change simulation comparison with bow waves. (**a**) The amplitude of 0.1 m; (**b**) the amplitude of 0.25 m; (**c**) the amplitude of 0.4 m.

The new parameters are listed in Table 4, and the simulation results are shown in Figure 15. Increasing the stiffness effectively reduces the dynamic stroke amplitude of

the suspension system. From the simulation results of the amplitude, the smaller the stiffness, the better the effect of buffering and absorbing the vertical movement, but from the perspective of the dynamic stroke amplitude, the impact condition with bow waves has higher requirements for the suspension system, which causes a greater dynamic stroke for the suspension system.

**Table 4.** Simulation parameters of stiffness value with bow waves.

| Stiffness Coefficient (N·m$^{-1}$) | Damping Coefficient (Ns·m$^{-1}$) | Gradient of Coefficient Change |
|---|---|---|
| 4500 | 441 | reference value |
| 5625 | 490 | +25% |
| 6750 | 538 | +50% |
| 7875 | 582 | +75% |
| 9000 | 622 | +100% |

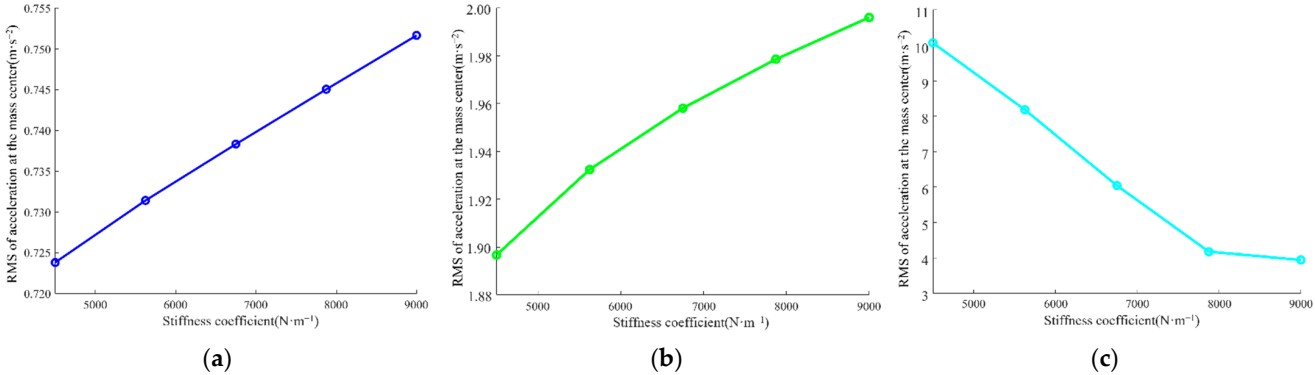

(**a**)          (**b**)          (**c**)

**Figure 15.** Stiffness change simulations comparison with bow waves. (**a**) The amplitude of 0.1 m; (**b**) the amplitude of 0.25 m; (**c**) the amplitude of 0.4 m.

### 4.3. Conclusions from the Study of Key Parameters

The simulation results show that the stiffness coefficients are 3375 N·m$^{-1}$ and 9000 N·m$^{-1}$, and the damping coefficients are 190 Ns·m$^{-1}$ and 622 Ns·m$^{-1}$. From the overall trend, the working condition with a large heave motion has little effect on the safety of the suspension system, and the stiffness coefficient and the damping coefficient are smaller, which is more helpful for the effect of buffering and absorbing vertical movement. However, under the working condition of bow waves, this causes a larger amplitude of the suspension system's dynamic stroke, so it requires a larger stiffness and damping coefficient of the suspension system to support the rescue platform and buffer the vertical movement. Specifically, under the condition of bow waves, it is necessary to increase 100% of the reference stiffness coefficient to ensure that the limited block is not touched with the impact of the large amplitude. When selecting the parameters of the actual suspension system in the prototype vessel, it is necessary to refer to this ultimate working condition for the calibration of the parameters.

To summarize, in contrast to the suspension system parameter selection of the automobile, the suspension system parameters of the rescue vessel require the selection of a larger stiffness and damping coefficient based on the reference selection due to the actual working conditions, and the stiffness and damping coefficient determined by the traditional automobile suspension parameter selection method is not enough for the rescue vessel.

## 5. Discussion

### 5.1. Effect of the Suspension System on Seakeeping

A ship seakeeping criterion is an index used to evaluate the seakeeping of a ship that refers to the limit index of whether the hull, personnel, or equipment systems on the ship

can operate normally and complete the corresponding tasks when the ship is disturbed by waves [39]. In this study, we mainly refer to the basic seakeeping criteria proposed by the Nordic Cooperative Research Program [40] for predicting the operational characteristics of ships, as shown in Table 5.

**Table 5.** Seakeeping criteria proposed by the Nordic Cooperative Research Program.

| Seakeeping Elements | Merchant Ships | Navy Vessels | Fast Small Craft |
|---|---|---|---|
| RMS of vertical acceleration at forebridge | 0.05–0.275 g | 0.275 g | 0.65 g |
| RMS of vertical acceleration at bridge | 0.15 g | 0.20 g | 0.275 g |
| RMS of lateral acceleration at bridge | 0.12 g | 0.1 g | 0.1 g |
| RMS of roll | 6.0° | 4.0° | 4.0° |
| Probability of slamming | 0.01–0.03 | 0.03 | 0.03 |
| Probability of deck wetness | 0.05 | 0.05 | 0.05 |

The prototype of a rescue vessel is a high-speed boat, so the seakeeping performance of the prototype is evaluated using the seakeeping criteria of small fast crafts. Only the RMS of the vertical acceleration, the RMS of the lateral acceleration, and the RMS of the roll of the main hull are considered. Tables 6 and 7 display the experimental data of the moderate waves and hard waves collected in the laboratory as the input conditions to calculate the seakeeping criteria with and without the suspension system. The results show that the seakeeping criteria of the rescue vessel meet the requirements of a high-speed boat under different sea conditions and after the suspension system is added. All the criteria values of the seakeeping performance of the rescue platform are lower than those of the rescue platform without the suspension system. Thus, it can be seen that the overall seakeeping performance can be improved by adding the suspension system.

**Table 6.** Seakeeping criteria values with moderate waves.

| Seakeeping Elements | With Suspension | Without Suspension | Criteria |
|---|---|---|---|
| RMS of vertical acceleration at bridge | 0.05 g | 0.057 g | 0.275 g |
| RMS of lateral acceleration at bridge | 0.052 g | 0.04 g | 0.1 g |
| RMS of roll | 1.559° | 1.569° | 4.0° |

**Table 7.** Seakeeping criteria values with hard waves.

| Seakeeping Elements | With Suspension | Without Suspension | Criteria |
|---|---|---|---|
| RMS of vertical acceleration at bridge | 0.124 g | 0.145 g | 0.275 g |
| RMS of lateral acceleration at bridge | 0.085 g | 0.077 g | 0.1 g |
| RMS of roll | 2.5543° | 2.5698° | 4.0° |

### 5.2. Effect of the Articulated System on Seakeeping

In the innovative design of the rescue vessel, a front arch mechanism is added between the rescue platform and the suspension system, and three spherical joints are used to connect the three parts. The main function of this innovative design is to make the pontoon move vertically relative to the rescue platform to a certain extent through the rotation of the front arch to reduce the impact of the pontoon movement on the rescue platform and achieve a certain wave adaptation effect. The spherical joint is of little help to the performance of the rescue vessel in the case of large heave motion, and it mainly has obvious wave adaptation to the impact of bow waves. To analyze the influence of the addition of the spherical joint on the performance of the rescue vessel, the impact inputs of the bow waves with different amplitudes and the influences with or without the spherical joint on the motion response of the rescue vessel are determined, as shown in Figure 16. The comparison of various seakeeping criteria values at the rescue platform with and

without spherical joints with the impact of bow waves with different amplitudes is shown in Figure 17.

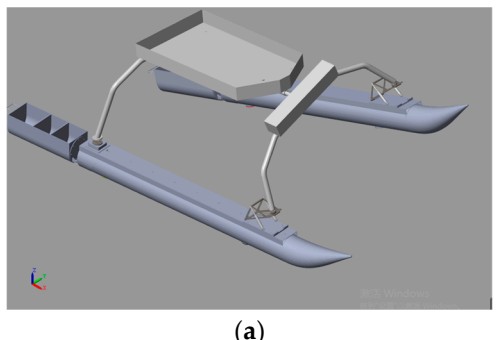 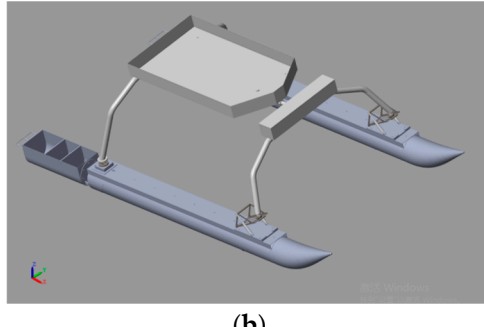

(**a**)          (**b**)

**Figure 16.** Simulation animation with or without the spherical joint. (**a**) With the spherical joint; (**b**) Without the spherical joint.

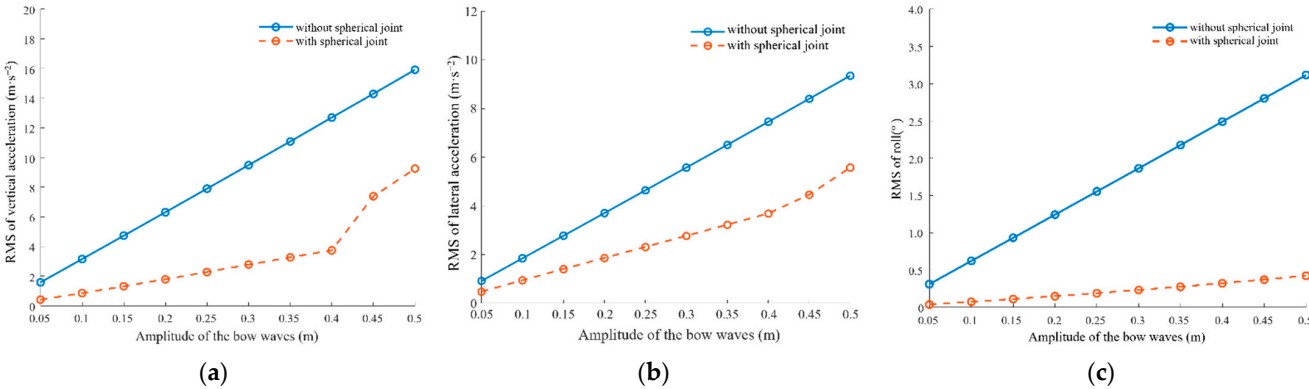

(**a**)       (**b**)       (**c**)

**Figure 17.** Seakeeping criteria values with and without spherical joint. (**a**) RMS of vertical acceleration; (**b**) RMS of lateral acceleration; (**c**) RMS of roll.

The simulation results show that after the addition of the spherical joint, the seakeeping performance of the rescue vessel is greatly improved under the condition of bow waves, from the perspective of the reduction of the vertical acceleration, lateral acceleration, and rolling angle. Hence, the rescue platform becomes stable and the operation at sea is more reliable. In the response results of the vertical acceleration and lateral acceleration, it is found that when the wave impact input amplitudes are 0.45 m and 0.5 m, there is a sudden change in the RMS value of the acceleration. Because the wave impact amplitude is too large, the rotation angle of the spherical joint reaches the maximum stroke. However, the RMS value of acceleration is still lower than that without the spherical joint.

Table 8 shows the influence of the spherical joint on the pitch motion and heave motion for different wave amplitudes in the simulation with bow waves. According to the simulation results, after the addition of the spherical joint, the pitch motion angle with the bow waves is increased to a certain extent. However, the percentage increase is not large, about 3%. Unlike the pitch motion, the heave motion amplitude with bow waves is greatly reduced by about 38% after the addition of the spherical joint. According to the comprehensive simulation results, the spherical joint can greatly reduce the heave motion amplitude under the condition of bow waves, but the increase in the pitch angle can be ignored. Furthermore, the rescue platform is more stable with bow waves, and to a certain extent, the wave adaptation effect is achieved.

**Table 8.** Pitch and heave response with and without spherical joint.

| Amplitude of Bow Waves(m) | RMS of Pitch (°) | | | Amplitude of Heave (m) | | |
|---|---|---|---|---|---|---|
| | With Spherical Joint | Without Spherical Joint | Percentage Increase | With Spherical Joint | Without Spherical Joint | Percentage Reduction |
| 0.05 | 0.3158 | 0.3097 | 1.97% | 0.0381 | 0.06256 | 39.10% |
| 0.1 | 0.6318 | 0.6094 | 3.68% | 0.07627 | 0.12513 | 39.05% |
| 0.15 | 0.9487 | 0.9294 | 2.08% | 0.11458 | 0.18766 | 38.94% |
| 0.2 | 1.267 | 1.24 | 2.18% | 0.153 | 0.2502 | 38.82% |
| 0.25 | 1.587 | 1.551 | 2.32% | 0.19183 | 0.3128 | 38.67% |
| 0.3 | 1.909 | 1.862 | 2.52% | 0.23107 | 0.3753 | 38.43% |
| 0.35 | 2.234 | 2.175 | 2.71% | 0.2706 | 0.4379 | 38.21% |
| 0.4 | 2.563 | 2.488 | 3.01% | 0.3108 | 0.5004 | 37.89% |
| 0.45 | 2.895 | 2.803 | 3.28% | 0.3516 | 0.563 | 37.55% |
| 0.5 | 3.233 | 3.119 | 3.66% | 0.3935 | 0.6256 | 37.10% |

## 6. Conclusions

The multi-body dynamic model of the rescue vessel is established based on Matlab/SimMechanics, and the correctness of the model is verified by the experiment with the prototype. The mean absolute error of the heave motion of the multi-body dynamic model is 0.01 m, and the mean absolute error of the roll and pitch motion is less than 1.82°. The model is correct and reliable, and the dynamic model of the rescue vessel can be used to predict the motion response.

The influence of the suspension system parameters on the acceleration response of the rescue platform under typical working conditions is analyzed, which provides a reference for the parameter selection of the suspension system.

The addition of the suspension system can show a good buffering and motion reduction effect under various working conditions. The most obvious effect is the reduction of the vertical acceleration at the rescue platform. The larger the amplitude of the wave impact, the more obvious the effect. In addition, the seakeeping criteria values at the rescue platform are reduced. However, the addition of the articulated system reduces the angle of roll and the amplitude of vertical motion at the rescue platform with bow waves. Therefore, the addition of the suspension system and the articulated system brings effective improvement to the seakeeping performance of the rescue vessel.

In the future, multivariate nonlinear regression (MNLR) combined with the multiobjective particle swarm optimization (MOPSO) method will be used to optimize the parameters of the suspension system [41], and comprehensive evaluation indicators will be used to measure the seakeeping performance of the rescue vessel [42]. In addition, the hydrodynamic problem between the flexible pontoons and the wave is also worth studying. The flexible pontoons have the function of suspension and vibration isolation similar to the tires of an automobile. In the future, the simplified panel method (sPM) method can be used to predict the motion response of the pontoons quickly [43].

**Author Contributions:** Conceptualization, J.L., X.B., Y.L., H.D., F.F., S.L., Z.L. and W.X.; methodology, J.L., X.B., Y.L., H.D. and W.X.; software, J.L., X.B. and W.X.; validation, J.L., X.B. and W.X.; formal analysis, J.L., X.B., Y.L., H.D., F.F., S.L., Z.L. and W.X.; investigation, J.L., X.B., Y.L., F.F., S.L. and Z.L.; resources, J.L., H.D. and W.X.; data curation, J.L. and W.X.; writing—original draft preparation, J.L., X.B. and W.X.; writing—review and editing, J.L., H.D. and W.X.; visualization, J.L., X.B. and W.X.; supervision, W.X.; project administration, J.L. and W.X.; funding acquisition, H.D. and W.X. All authors have read and agreed to the published version of the manuscript.

**Funding:** This research was funded by the National Natural Science Foundation of China, grant number 51905066 and 52075065.

**Institutional Review Board Statement:** Not applicable.

**Informed Consent Statement:** Not applicable.

**Data Availability Statement:** Not applicable.

**Acknowledgments:** The authors acknowledge the technical support from the Dalian Maritime University School of Marine Engineering staff. The authors appreciate the feedback of the reviewers and the editors on this submission.

**Conflicts of Interest:** The authors declare no conflict of interest. The funders had no role in the design of the study; in the collection, analyses, or interpretation of data; in the writing of the manuscript, or in the decision to publish the results.

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
