# Peer review of "Investigation of a Cabin Suspended and Articulated Rescue Vessel in Terms of Motion Reduction"

_jmse, doi:10.3390/jmse10121966_

Round 1
Reviewer 1 Report
Please read the attachment. Thank you.

Reviewer 2 Report
The novelty of the study should be clearly addressed in the paper.
Comparison with the other related works should be added in the manuscript.
English of the paper can be improved.
Reviewer 3 Report
This paper presents the design of a suspended and articulated cabin to solve the problem of small rescue vessels. The multi-body dynamic model of the whole vessel based on SimMechanics was built, and the prototype was tested on the water.
I believe this topic is interesting, but this paper must be re-outlined in order to clarify the main ideas and the contribution. Some sections must be added or improved:
1) It is important to realize a comparative analysis of this proposal with other related works in order to clarify the contribution. I suggest including a comparative analysis of other related works, in order to compare and identify the main contributions and expose the differences between this work and other seminal works
2) A Evaluation/Discussion section is missing. In this section, discussion paragraphs should be introduced. In these paragraphs, the main contributions of the results presented should be emphasized.
3) The conclusions section is not good and the authors have not properly commented on all outcomes of this work. I would like to suggest including more future work due to the relevance of this work.
4) The authors need to include more high-quality and high-cited references. In this work, the quality of the references is acceptable but some works were published in proceedings.
5) The English must be improved. There are problems with style and grammar that would be best resolved through proofreading and copy editing by another proficient English speaker.
In its present form, it is not clear what is the innovation and the “academic” value of this proposal.
Round 2
Reviewer 3 Report
This paper has been improved in a significant way. The authors have addressed all reviewer's comments. This paper can be accepted for publication in its present form.